# Severe Atrophy of the Ipsilateral Psoas Muscle Associated with Hip Osteoarthritis and Spinal Stenosis—A Case Report

**DOI:** 10.3390/medicina57010073

**Published:** 2021-01-15

**Authors:** Byeongcheol Lee, Sang Eun Lee, Yong Han Kim, Jae Hong Park, Ki Hwa Lee, Eunsu Kang, Sehun Kim, Nakyung Lee, Daeseok Oh

**Affiliations:** Department of Anesthesia & Pain Medicine, Inje University Haeundae Paik Hospital, Busan 612-896, Korea; h00543@paik.ac.kr (B.L.); painlee@paik.ac.kr (S.E.L.); h00111@paik.ac.kr (Y.H.K.); h00150@paik.ac.kr (J.H.P.); tedy333@paik.ac.kr (K.H.L.); h00347@paik.ac.kr (E.K.); shottherbz@paik.ac.kr (S.K.); H80550@paik.ac.kr (N.L.)

**Keywords:** psoas muscles, muscular atrophy, hip osteoarthritis, radiating pain

## Abstract

Pathology of the lumbar spine and hip joint can commonly coexist in the elderly. Anterior and lateral leg pain as symptoms of hip osteoarthritis and spinal stenosis can closely resemble each other, with only subtle differences in both history and physical examinations. It is not easy to identify the origin of this kind of hip pain. The possibility of hip osteoarthritis should not be underestimated, as this could lead to an incorrect diagnosis and inappropriate spinal surgery. We report the case of a 54-year-old female with chronic right anterior and lateral leg pain who did not respond to repeated spinal blocks based on lumbar MRI, but in whom hip osteoarthritis was considered since severe atrophy of the ipsilateral psoas muscle was identified. We suggest that severe psoas muscle atrophy can be a clinical clue to identify hip osteoarthritis and is related to lower extremity pain, even if there is a coexisting lumbar spine pathology.

## 1. Introduction

Hip osteoarthritis and lumbar spinal stenosis generally occur in elderly patients. Spinal stenosis is a condition characterized by reduced spinal canal and compression of neural elements secondary to degenerative changes in the spinal canal. Neurogenic claudication, which is defined by pain or radiculopathy in the buttocks or lower extremities, occurs as a result of ischemia due to venous congestion and arterial hypertension around nerve root or mechanical compression of nerve roots [1]. The distribution of pain in the lower extremities depends on the area of stenosis. Pain caused by hip osteoarthritis is attributed to the subchondral bone, periosteum, synovium, ligaments, and the joint capsule, which are all richly innervated and carry nerve endings [2]. The hip pain and stiffness subsequently reduce patient mobility [3]. Chronic joint pain is also associated with central sensitization, which can lead to referred pain and even tenderness distant from the affected joint [4]. The lower limb distribution of the pain in hip joint osteoarthritis and spinal stenosis using body image maps showed very similar patterns, with only subtle differences [5]. In patients with both lumbar spinal stenosis and hip joint osteoarthritis, a definitive and differential diagnosis and identifying the anatomic source of lower leg pain is often more complicated [6,7,8,9]. The origin of pain is difficult to establish if patients with spine pathology and degeneration of hip joint manifest pain in the lateral aspect of the lower leg [6]. However, hip joint osteoarthritis itself is usually an uncomplicated diagnosis that should not be overlooked to avoid costly and potentially hazardous diagnosis, as well as needless spinal surgery [7]. The case report introduces a patient with anterior and lateral thigh pain who failed to respond to repeated spinal blocks based on lumbar MRI findings, but hip joint osteoarthritis was suspected since a significant atrophy of the ipsilateral psoas muscle was identified. 

## 2. Case

The patient was a 54-year-old female with chronic severe right anterior and lateral leg pain with radiation above the knee and mild low back pain. Her pain score was 8/10 in the numeric rating scale (NRS). Prior to visiting our clinic, she had been diagnosed with multilevel lower lumbar degenerative changes, including a bulging disc and central canal stenosis at L3-4 and L4-5 on lumbar MRI. She received fluoroscopic-guided transforaminal epidural blocks repeatedly at the local spine center. However, the spinal blocks were not effective for her anterior thigh pain, and so lumbar spine surgery was planned. However, she refused to have lumbar surgery and visited our pain clinic. In her history, she was an officer and worked seated for long periods. She usually crossed her legs while sitting until the pain got worse. Although the patient could not remember the duration of the disease, she said that there was intermittent pain for about 25 years since childbirth and the pain had gradually exacerbated recently. There was no remarkable medical history or antecedent trauma. The patient continued to walk with a limp, which worsened with walking up a steep incline or climbing stairs. Her pain also was aggravated by movement and walking. Physical examination revealed mild tenderness over the right groin area. The patient was unable to fully extend her right hip joint to neutral in the supine position because it reproduced anterior thigh and low back pain and a sensation of tightness. The leg was not completely straightened. The Thomas test of the right side was positive. Her right hip joint showed a flexion contracture of about 15° due to limited range of motion in the supine position. The patient stated that she always slept with a pillow under her right knee or on the side. The Patrick and Ganslene test results were also positive on the right leg. The straight leg raising test was 80/80. Neurologic examinations revealed weakness of the hip flexor (grade 4/5 on the right side, grade 5/5 on the left side based on the MRC scale). We identified a weakness in the right hip flexors compared to the contralateral leg based on a resisted flexion test of the hip joint. No sensory changes were detected using pinprick and light touch sensation in the anterior and lateral aspects of the lower leg. No reflex abnormality of the legs was detected. 

Our clinical suspicion was that her right thigh pain was caused by other problems. Therefore, we evaluated her lumbar MRI taken at the local spine center, and were able to identify severe atrophy of the right psoas muscle besides degenerative spine (Figure 1 and Figure 2). A plain radiograph of the hip joint was evaluated additionally, and right hip joint osteoarthritis with narrowing of the joint space was identified (Figure 3). We performed ultrasound-guided intramuscular injection of the right psoas muscle with 10 ccs of 0.25% lidocaine for muscle tightness and nerve entrapment. Her pain improved on NRS 5~6/10, but she still complained of discomfort. Then, we performed ultrasound-guided, diagnostic, intra-articular injection of the right hip with 5 ccs of 0.5% lidocaine mixed with 10 mg of triamcinolone. At the time of a follow-up visit after one week, the patient reported that the pain level was 3/10 on NRS and that she had improved function, enabling her to straighten her leg. However, the flexion contracture of the right hip joint was not resolved completely. After two weeks, the anterior and lateral leg pain aggravated gradually, so the patient was referred to the Department of Hip Orthopedic Surgery for the evaluation of hip surgery. Informed consent for the publication was approved by the patient. 

## 3. Discussion

The psoas muscle, an important hip flexor, is the largest muscle in the lower lumbar spine and is innervated by the lumbar spinal nerve roots L2-L4 [10]. It has been suggested that weakness of the iliopsoas muscle is related to a neurogenic compression of the L2-L4 spinal roots and may be a clinical feature of lumbar spinal radiculopathy [11]. However, a systemic review of previous studies has shown conflicting results about the effects of unilateral lumbar pathology on psoas muscle size [12]. A radiologic study to assess a cross-sectional area with a lumbar spine MRI reports that atrophy of the paraspinal and psoas ipsilateral muscle develops in patients with persistent unilateral back pain or degenerative disc disease [13,14,15]. Conversely, some authors have reported that the psoas muscle on MRI imaging is not influenced by unilateral lumbar radiculopathy [16,17,18]. Chon et al. [16] suggest that denervation atrophy may not occur in the psoas muscle, as it acts as a hip flexor and is not innervated directly by the dorsal ramus of the medial branch, which is different than the multifidus muscle or the erector spinae.

Hip joint degenerative disease can also be associated with a reduction in psoas muscle strength. This muscular weakness is related to pain-mediated psoas muscle disuse atrophy [19]. The anterior hip joint capsule is innervated by the articular branches of femoral and obturator nerves that exit through the psoas muscle [20]. Rasch et al. [21] report that muscular weakness around the affected joint is also a clinical feature in lower limb osteoarthritis and that hip flexion strength is reduced. They report that the cross-sectional area of the ipsilateral psoas muscle is reduced 19% concretely in hip osteoarthritis relative to the limb of healthy patients. Although there are real debates regarding the pathogenesis of muscle weakness and subsequent atrophy in hip osteoarthritis, muscle weakness can be a cause of hip osteoarthritis, and osteoarthritis can also be a cause of muscle weakness [22]. In the advanced stage, psoas muscle pathology itself can cause pain in the lower back, pelvis, and the femoral region [23].

In our case, a cross-sectional area of both psoas muscles was measured via computer-assisted analysis using ImageJ software (Version 1.53, https://imagej.nih.gov/ij/, National Institutes of Health). The cross-sectional area of the right psoas muscle was about 0.55-fold (4117/7460 pixel counting) smaller than that of the contralateral side at the L3-4 level. The mean signal intensity of the selected areas was used as an indicator of fatty infiltration, with high signal intensity reflecting greater fatty infiltration, and was calculated using ImageJ software [24]. The results showed greater fat infiltration on the right psoas muscle compared to the left side (Figure 1 and Figure 2). A significant muscle atrophy of ipsilateral psoas muscle was identified.

Although our patient had spinal stenosis on MRI at the L3-4 and the L4-5 levels, repeated spinal blocks done under fluoroscopy did not resolve her lower leg pain. We considered it necessary to exclude the presence of ipsilateral hip osteoarthritis because degenerative disease of the hip can also be associated with a reduction in iliopsoas muscle strength. The degree of ipsilateral psoas muscle atrophy was remarkably greater than that of the other paraspinal muscles, namely the multifidus and erector muscle. Additionally, on MR imaging, there was central pressure on the cord. The pressure on the left was greater, but atrophy in the right psoas muscle was extreme.

We considered her left hip joint osteoarthritis to be the main cause of her anterior and lateral thigh pain after we had identified severe ipsilateral psoas muscle atrophy, although we could not establish the exact mechanism of our patient’s muscle atrophy on physical examination. The difficulty of diagnosing the origin of ipsilateral lower leg pain in patients with both lumbar spinal stenosis and hip osteoarthritis has been described [6,7,8,9]. However, factors that demonstrate a good efficacy in detecting this kind of pain’s origin have not been established completely. More investigations to resolve these diagnostic difficulties will be done in future studies. We suggest that the severe psoas atrophy that we saw on lumbar MRI in patient with spinal stenosis led us suspect that her anterior and lateral leg pain was caused by ipsilateral hip osteoarthritis.

## 4. Conclusions

This is a unique clinical report in which both lumbar spinal stenosis and hip osteoarthritis were simultaneously present in a patient with severe ipsilateral psoas atrophy. The presence of severe atrophy of ipsilateral psoas muscle is a pathognomonic of hip joint osteoarthritis and is associated with anterior and lateral leg pain, despite coexisting lumbar spinal pathology. We emphasize that this finding has discriminating value and can be used to aid in differential diagnosis. Further controlled trials are needed to evaluate the validity of the clinical finding.

## Figures and Tables

**Figure 1 medicina-57-00073-f001:**
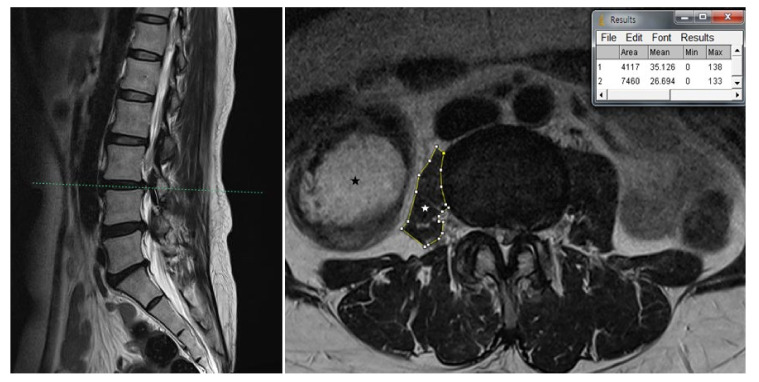
Scout view of the axial T2-weighted lumbar MRI shows distinct atrophy of right psoas muscle compared to the left side in the L3-4 intervertebral space. The MRI reveals diffuse disc bulge and central canal stenosis with bilateral facet joint arthropathy and ligamentum hypertrophy in L3-4. Cross-sectional area (CSA) analysis and signal intensities were calculated using ImageJ software. The CSA of both psoas muscles was measured by constructing polygons around the outer margins of muscle. The CSA of the right psoas muscle was about 0.55-fold (4117/7460 pixel counting) smaller than that of the left side. The mean signal intensities directly obtained from the area were 35.126 (area 1) and 26.694 (area 2): (1) CSA of the right psoas muscle; (2) CSA of the left psoas muscle. White asterisk indicates right psoas muscle. Black asterisk indicates fecal content in the colon.

**Figure 2 medicina-57-00073-f002:**
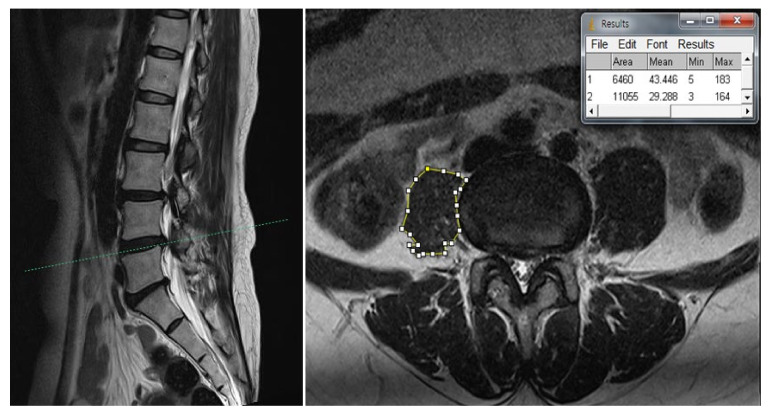
Scout view of the axial T2-weighted lumbar MRI shows distinct atrophy of right psoas muscle compared to the left side in the L4-5 intervertebral space. Cross-sectional area (CSA) of the right psoas muscle was about 0.58-fold (6460/11,055 pixel counting) smaller than that of the left side. The mean signal intensities were 43.446 (area 1) and 29.288 (area 2): (1) CSA of the right psoas muscle; (2) CSA of the left psoas muscle.

**Figure 3 medicina-57-00073-f003:**
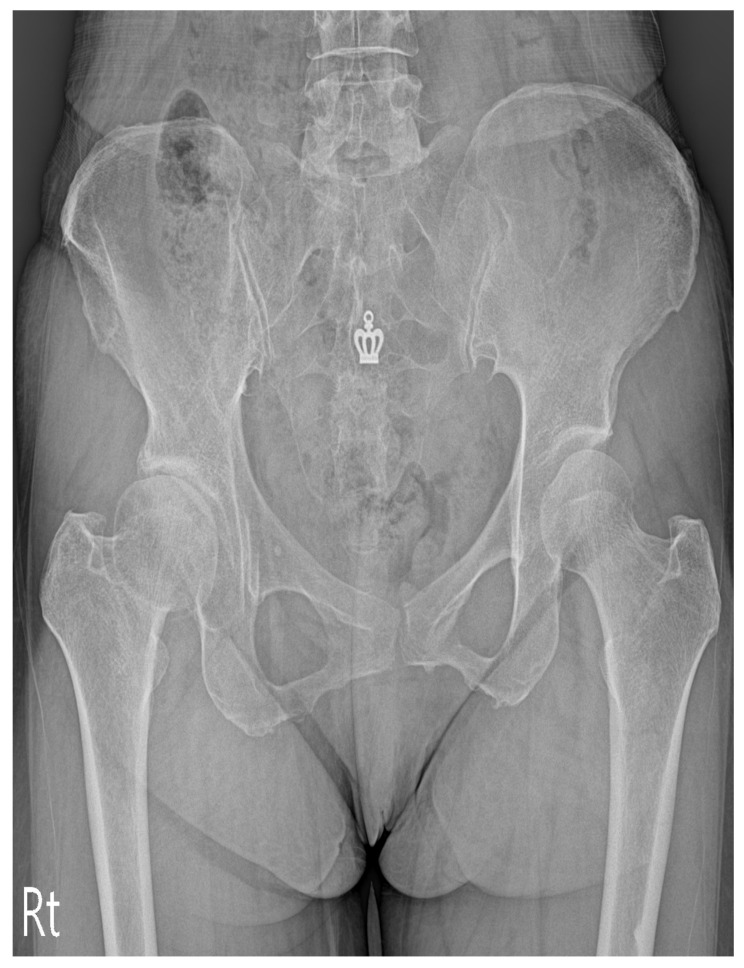
Anteroposterior view of the pelvis in the supine position shows decreased right-side hip joint space, subchondral sclerosis.

## Data Availability

The data presented in this study are available on request from the corresponding author.

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
