# Peer review of "Severe Atrophy of the Ipsilateral Psoas Muscle Associated with Hip Osteoarthritis and Spinal Stenosis—A Case Report"

_medicina, 2021, doi:10.3390/medicina57010073_

Round 1
Reviewer 1 Report
In this case report, authors described a 54 years old Patient with chronic right anterior and lateral leg pain. In this Patient, both lumbar spinal stenosis and hip osteoarthritis with severe ipsilateral psoas muscle atrophy were present at the same time. Ipsilateral psoas muscle atrophy can be a clinical clue to identify hip joint osteoarthritis and is related to anterior and lateral leg pain, even if there is a coexisting lumbar spine pathology.
The main weakness of this study is the very poor quality of the radiological images.
The authors reported only three figures: 2 MRIs and 1 Xray.
Regarding MRI images, they inserted only one sequence (T2w) on axial and sagittal plane.
I would advise the authors to add more images, including different sequences, in particular STIR, that is essential for the evaluation of muscle edema, different acquisition planes (coronal plane could be very useful to study ileo-psoas), and different levels.
The captions are too short and do not properly explain the figures.
Furthermore, they do not show how they segmented and measured the ileo-psoas, which might be interesting to evaluate.
The authors claim to have measured the cross-sectional area of both ileopsoas but do not specify at what level they did so.
Finally, in the only axial image inserted it is possible to notice an enormous hypoerintensity of the left kidney, which was not described in the caption.
The scientific relevance of this case report is not adequate and does not bring any new information on both clinical and radiological management of these patients.
Author Response
Response to Reviewer 1 Comments
Thank you for your kind comments. We got a lot of help from you in revising my manuscript. We considered your opinions as much as possible. We appreciate your reviewing and look forward to receiving further response.
In this case report, authors described a 54 years old Patient with chronic right anterior and lateral leg pain. In this Patient, both lumbar spinal stenosis and hip osteoarthritis with severe ipsilateral psoas muscle atrophy were present at the same time. Ipsilateral psoas muscle atrophy can be a clinical clue to identify hip joint osteoarthritis and is related to anterior and lateral leg pain, even if there is a coexisting lumbar spine pathology.
The main weakness of this study is the very poor quality of the radiological images.
The authors reported only three figures: 2 MRIs and 1 Xray.
Regarding MRI images, they inserted only one sequence (T2w) on axial and sagittal plane. I would advise the authors to add more images, including different sequences, in particular STIR, that is essential for the evaluation of muscle edema, different acquisition planes (coronal plane could be very useful to study ileo-psoas), and different levels.
Response: We agree with your opinion. Unfortunately, we did not have coronal plane and STIR images because her MRI examination was taken at other hospital prior to visiting our clinic. So, we could not identify these images.
Instead, we modified the figure 1 and added figures (T1w image, axial image at L4-5 level). And we additionally measured the signal intensity to evaluate muscle quality by ImageJ
Page 3, Figure 1, 2
The captions are too short and do not properly explain the figures.
Response: This flaw has been corrected.
Page 3-4, Figure 1, 2, 3
Furthermore, they do not show how they segmented and measured the ileo-psoas, which might be interesting to evaluate.
The authors claim to have measured the cross-sectional area of both ileopsoas but do not specify at what level they did so.
Response: We described it correctly and added explanations.
Page 3, Figure 1, 2
Finally, in the only axial image inserted it is possible to notice an enormous hypoerintensity of the left kidney, which was not described in the caption.
Response: We reviewed MRI images and confirmed that the enormous hypointensity part was feces in the bowel.
The scientific relevance of this case report is not adequate and does not bring any new information on both clinical and radiological management of these patients.
Response: We emphasize that this case is a unique clinical report in which both lumbar spinal stenosis and hip osteoarthritis were simultaneously present in a patient with severe ipsilateral psoas atrophy and The presence of severe atrophy of ipsilateral psoas muscle is a pathognomonic of hip joint osteoarthritis and is associated with anterior and lateral leg pain, despite coexisting lumbar spinal pathology.
Page 5, Lines 155-159

Reviewer 2 Report
Abstract
The last part of the abstract should explain briefly the findings of the study, otherwise readers may find the article uninteresting.
Introduction
Introduction section is quite poor, it is just the same of abstract. Authors should add some citation in order to clarify the relationship between pain and lumbar spinal stenosis, pain and hip osteoarthritis, and then specify that both symptoms can be very similar, otherwise it looks like elder patients can have commonly hip OA or LSS which determinate the same symptoms. I think it is wrong because one occurs with radicular problems, the other with mechanical problems, involving into a different type of pain.
Case
Line 36 Which pain scale was used? VAS? CR10? 6-20? State it.
Line 47 Why authors reported only Patrick’s test? Have not they done any other tests? Lasegue, Valsalva, Wassermann, reflexes, motor and sensor alterations? How can readers be sure that the pain does not come from lumbar nerve roots?
Line 50 What kind of ROM limitation is referred to? We do not know if the patient had limitation in flexion too, or more generally, which the ROM of hip joint is. If there is hip OA, there should be also a limitation in passive flexion, right?
Line 51-52 The weakness of hip flexors have not a reference scale like MRC. So readers cannot understand if there was a small or huge difference between both legs.
Line 52-53 Readers cannot be sure that there were no sensory changes since authors reported only Patrick’s test, which is not a test for nerve root sensitivity.
Line 57-59 Authors reported an intramuscular injection of the right psoas, but seems like it did not change the perceived pain, so, why authors are supposing that psoas weakness can be associated with anterior and lateral leg pain?
Line 63 Early authors did not report the hip joint ROM so now it is hard to understand the increase of the hip joint ROM.
Figure 2. Is the radiography done supine or standing? State it.
Discussion
Line 77 As reported here, authors are citing the relationship with L2-L4 neurogenic compression, I still wonder why early no tests for radiculopathy were made.
Conclusions
Conclusions seem to be poor of contents. Even if data did not show a great relationship between atrophy of the psoas muscle with hip OA and LSS, authors can still highlight the need of future investigations. Please add limitations of the study and please provide a clear message of the importance of this paper in the scientific community.
References
All the references have the numbers doubled, delete the ones hand wrote and just leave the numbered list.
Line 169 There is a 16. without reference, delete it.
Author Response
Response to Reviewer 2 Comments
Thank you for your kind comments. We got a lot of help from you in revising my manuscript. We considered your opinions as much as possible. We did our best to revise my manuscript. We appreciate your reviewing and look forward to receiving further response.
Abstract
The last part of the abstract should explain briefly the findings of the study, otherwise readers may find the article uninteresting.
Response: We described it correctly and added explanations
We suggest that severe psoas muscle atrophy can be a clinical clue to identify hip osteoarthritis and is related to lower extremity pain, even if there is a coexisting lumbar spine pathology.
Page 1, Lines 21-23
Introduction
Introduction section is quite poor, it is just the same of abstract. Authors should add some citation in order to clarify the relationship between pain and lumbar spinal stenosis, pain and hip osteoarthritis, and then specify that both symptoms can be very similar, otherwise it looks like elder patients can have commonly hip OA or LSS which determinate the same symptoms. I think it is wrong because one occurs with radicular problems, the other with mechanical problems, involving into a different type of pain.
Response: We described it correctly and added some citations.
- Pain and spinal stenosis
- Pain and hip osteoarthritis
- Difficulty of diagnosing the origin of lower leg pain
- Purpose of this case report
Page 1-2, Lines 27-, 47
Case
Line 36 Which pain scale was used? VAS? CR10? 6-20? State it.
Response: We described it correctly. The pain scale was NRS.
Page 2, Line 50
Line 47 Why authors reported only Patrick’s test? Have not they done any other tests? Lasegue, Valsalva, Wassermann, reflexes, motor and sensor alterations? How can readers be sure that the pain does not come from lumbar nerve roots?
Response: We added the findings of physical examination.
SLRT, Patrick and Ganslene test, Thomas test, and neurologic test
Page 2, Lines 61-72
Line 50 What kind of ROM limitation is referred to? We do not know if the patient had limitation in flexion too, or more generally, which the ROM of hip joint is. If there is hip OA, there should be also a limitation in passive flexion, right?
Response: I agree with your opinion. We added the findings of ROM limitation. Her right hip joint showed a flexion contracture of about 15° due to limited range of motion in the supine position.
Page 2, Lines 65-66
Line 51-52 The weakness of hip flexors have not a reference scale like MRC. So readers cannot understand if there was a small or huge difference between both legs.
Response: We described it correctly. We expressed MRC scale.
Page 2, Line 69
Line 52-53 Readers cannot be sure that there were no sensory changes since authors reported only Patrick’s test, which is not a test for nerve root sensitivity.
Response: We described it correctly and added physical exam and neurologic test
Page 2, Lines 68-72
Line 57-59 Authors reported an intramuscular injection of the right psoas, but seems like it did not change the perceived pain, so, why authors are supposing that psoas weakness can be associated with anterior and lateral leg pain?
Response: We revised this statement.
Her pain improved on NRS 5~6/10 after intramuscular injection, but she still complained of discomfort. We could identify the mild symptom improvement.
We judged that it is difficult to solve completely joint problem with intramuscular injection alone. We don’t know that the pathogenesis of muscle weakness and subsequent atrophy in hip osteoarthritis. But we think that the problems associated severe psoas atrophy and hip joint osteoarthritis contributes to her pain and functional disorders.
Page 2, Lines 77-79
Line 63 Early authors did not report the hip joint ROM so now it is hard to understand the increase of the hip joint ROM.
Response: We added limitation of ROM.
Page 2, Lines 65-66
Figure 2. Is the radiography done supine or standing? State it.
Response: We described it correctly. In the supine position.
Page 4, Figure 3
Discussion
Line 77 As reported here, authors are citing the relationship with L2-L4 neurogenic compression, I still wonder why early no tests for radiculopathy were made.
Response: We agree with your point. We added the neurologic findings.
We checked sensory with pinprick and light touch sensation and motor check. There was no sensory deficit. SLRT is negative.
Page2, Lines 72-73
Conclusions
Conclusions seem to be poor of contents. Even if data did not show a great relationship between atrophy of the psoas muscle with hip OA and LSS, authors can still highlight the need of future investigations. Please add limitations of the study and please provide a clear message of the importance of this paper in the scientific community.
Response: We revised the conclusion and added limitations.
The presence of severe atrophy of ipsilateral psoas muscle is a pathognomonic of hip joint osteoarthritis and is associated with anterior and lateral leg pain, despite coexisting lumbar spinal pathology. This finding is of primary importance in differential diagnosis. Further controlled trials are needed to evaluate the validity of the clinical findings.
Page 5, 155-159
References
All the references have the numbers doubled, delete the ones hand wrote and just leave the numbered list.
Line 169 There is a 16. without reference, delete it.
Response: We revised it.
Page 6~8, References

Round 2
Reviewer 1 Report
I thank the authors for their valuable review, which considerably improves the paper. There are still some minor concerns:
- it remains unclear if you use T1w or T2w sequences to segment the ileo-psoas, it seems T2 from the images showed, but it would be better to specify it.
- you referred to feces in the bowel as hypointensity in Figure 1, did you mean hyerintensity? I imagine so and I that it should be described in the legend.
Author Response
Thank you for your kind response.
I thank the authors for their valuable review, which considerably improves the paper. There are still some minor concerns:
- it remains unclear if you use T1w or T2w sequences to segment the ileo-psoas, it seems T2 from the images showed, but it would be better to specify it.
Response: We removed T2 weighted image in the figure 2.
- you referred to feces in the bowel as hypointensity in Figure 1, did you mean hyerintensity? I imagine so and I that it should be described in the legend.
Response: We described it in the legend. Figure 1.

Reviewer 2 Report
Fix the text editing and images (size, allineation), the radiography is reported twice.
I would suggest to extend a bit the Conclusions. This, unlike the first version, seems to be excessively synthesized and does not allow us to understand the final result of the study.
Author Response
Thank you for your kind response.
Fix the text editing and images (size, allineation), the radiography is reported twice.
Response: We removed same images and modified Figure 2.
I would suggest to extend a bit the Conclusions. This, unlike the first version, seems to be excessively synthesized and does not allow us to understand the final result of the study.
Response: We modified conclusion. We mentioned the diagnostic value.
Page 5, Lines 154-158
